# TGPO: Efficient Policy Optimization through Sequence Anchor and Information Gating

**Hang Ding**[1]  **Dongqi Liu**[2]  **Qiming Feng**[3]  **Jian Li**[4 5]  **Tong Lei**[4]  **Jiafu Wu**[5]  **Shuo Wang**[5]  **Jiangning Zhang**[5]
**Chengjie Wang**[5]  **Yabiao Wang**[6 5]

## Abstract

Reinforcement learning from verifiable rewards (RLVR) has become an important paradigm for enhancing the reasoning capabilities of large language models, while it also involves a persistent tradeoff between optimization stability and learning efficiency. Token-level importance weighting supports fine-grained credit assignment, but it often introduces high variance and unstable parameter updates, whereas sequence-level optimization provides more stable learning dynamics while failing to fully exploit informative local signals. We introduce **T**rust-**G**ated **P**olicy **O**ptimization (TGPO), an efficient policy optimization framework that integrates two complementary mechanisms, namely *sequence anchors* and *information gates*. TGPO aligns token-wise updates with a stable sequence-level reference, which reduces the influence of extreme local likelihood fluctuations on the gradient, and a trust-based information gate adaptively modulates the contribution of token-level signals. By retaining and reweighting gradients from imperfect trajectories rather than excluding them, TGPO improves gradient utilization and sample efficiency while maintaining stable optimization behavior. Empirical results across seven mathematical reasoning datasets and multiple model scales show that TGPO consistently enhances learning efficiency and overall performance in outcome-supervised reinforcement learning settings.

[1]Shanghai Jiao Tong University  [2]Saarland University  [3]Fudan University  [4]Nanjing University  [5]Tencent Youtu Lab  [6]Zhejiang University. Correspondence to: Yabiao Wang <yabiaowang@zju.edu.cn>.

*Proceedings of the $43^{rd}$ International Conference on Machine Learning*, Seoul, South Korea. PMLR 306, 2026. Copyright 2026 by the author(s).

## 1. Introduction

Reinforcement Learning from Verifiable Rewards (RLVR) has emerged as an important paradigm for post-training large language models (LLMs), particularly in outcome supervised settings where feedback is provided at the response level (DeepSeek-AI et al., 2025; Yu et al., 2025). A prominent line of work builds on Group Relative Policy Optimization (GRPO) and its extensions, which rely on group-wise normalization to support stable and scalable optimization (Yu et al., 2025; Zeng et al., 2025; He et al., 2025; Chen et al., 2025; An et al., 2025; Wang et al., 2025b; Zhang et al., 2025). Although these methods demonstrate strong performance, group-based optimization continues to face a structural tradeoff between *sequence-level stability* and *token-level efficiency*. In particular, mechanisms that favor stable sequence-level updates tend to weaken the learning signal available at finer granularity, while token-wise importance weighting is often associated with elevated variance and unstable parameter updates(Wang et al., 2025a; Sheng et al., 2025b).

To better understand this tradeoff, we revisit the role of importance sampling (IS) (Precup et al., 2000) in outcome-supervised RL. While IS ratios are formally introduced as distribution-correction factors, under response-level rewards they effectively act as token-level training weights, determining how much each token contributes to the gradient update without knowing which tokens are actually responsible for the final outcome. This creates a mismatch between token-level likelihood correction and sequence-level supervision. In practice, if token-level ratios are left unconstrained, a small number of tokens with extreme importance ratios can dominate the update, leading to unstable KL behavior, rapid entropy reduction, and premature convergence. However, directly clipping these ratios introduces a different problem: local clipping errors may propagate through the sequence-level update, while initially low-probability but potentially important policy changes can be suppressed, removing useful gradient information and leading to inefficient data usage.

An alternative line of work addresses these issues through sequence-level importance sampling, as exemplified by

GSPO, which assigns a single importance ratio to each response. This design improves robustness by construction, but introduces two related efficiency limitations. First, GSPO is prone to gradient underutilization caused by conservative clipping. Because the sequence-level ratio is sensitive to deviations at any single position, a likelihood shift at one token can move the entire response outside the clipping range, nullifying useful gradients and leading to insufficient effective signal, slow reward improvement, and low sample utilization (Sheng et al., 2025b). Moreover, when many trajectories are truncated in this all-or-nothing manner, the resulting gradient signal can become sparse and high-variance, producing gradient-norm spikes and accelerating entropy collapse. Second, sequence-level weighting amplifies the flat credit assignment problem. Reasoning trajectories are typically heterogeneous, combining a small number of decisive reasoning steps with a large amount of routine or template-like content. Uniformly weighting all tokens spreads the learning signal too evenly, which limits the model's ability to focus on the critical decisions that determine the reward.

To address this stability efficiency tradeoff, we propose **Trust-Gated Policy Optimization (TGPO)**, a simple and grounded modification that combines sequence-level robustness with adaptive token-level learning. Instead of treating all token-wise importance ratios as equally reliable, TGPO interprets them through a notion of trust. The method constructs anchored updates that align token-level learning signals with a stable sequence-level reference, while an explicit information gate regulates the extent to which fine-grained token information contributes to the update. Each response is summarized using robust statistics that capture both the global drift of the sequence under the updated policy and its internal consistency across tokens.

These signals jointly determine a trust score that governs the learning behavior. Responses that exhibit moderate global drift and coherent internal structure allow more expressive token-level updates, whereas responses with large deviations or high internal variability are updated in a more conservative manner. This continuous interpolation preserves the stability properties associated with sequence-based methods while enabling the extraction of informative gradients from imperfect responses, rather than discarding them or allowing outlier tokens to dominate optimization. Empirically, TGPO leads to smoother optimization dynamics, reduced KL volatility, and improved data efficiency, providing a practical framework for trust-aware importance weighting in outcome supervised reinforcement learning. **Our main contributions are summarized as follows:**

- We provide a detailed analysis of token-level and sequence-level importance sampling in outcome-supervised, group-normalized RL, and show that these design choices lead to variance amplification, inefficient gradient utilization, or unusual gradient norm spikes that obscures decisive learning signals.

- We introduce Trust-Gated Policy Optimization, which integrates a robust *sequence anchor* with a *dual-trust information gate* derived from global sequence drift and within-sequence consistency, enabling adaptive control over the granularity of policy updates.

- We empirically show that TGPO leads to more stable optimization behavior and more effective use of training data, which in turn supports smoother learning dynamics and improved final performance across tasks.

## 2. Background

Reinforcement learning for large language model post-training has moved from Proximal Policy Optimization (PPO) (Schulman et al., 2017) toward critic-free reinforcement learning with verifiable rewards, with Group Relative Policy Optimization (GRPO) (Shao et al., 2024) as a representative approach. GRPO estimates advantages through group-wise relative comparisons among multiple responses to the same prompt, avoiding explicit value-function learning. Several works extend GRPO to improve training stability and efficiency. DAPO (Yu et al., 2025) applies dynamic sampling to reduce entropy collapse and exclude low-information groups, while GSPO (Zheng et al., 2025) shifts optimization from the token level to the sequence level to better align updates with response-level rewards.

Further variants modify aggregation and weighting strategies. GMPO and GTPO (Zhao et al., 2025; Simoni et al., 2025) introduce entropy-aware or geometric aggregation, and GRPO-S (Tan et al., 2025) uses sequence-ratio weighting to reduce sensitivity to outliers and sparse rewards. DCPO (Yang et al., 2025) instead adapts token-level clipping to improve exploration and utilize low-quality rollouts. Related directions focus on reward normalization and data construction. BNPO (Xiao et al., 2025) normalizes advantage scales, while curriculum-style reasoning data, such as Open-Reasoner-Zero (Hu et al., 2025) structures training by reasoning difficulty. Strong GRPO-based baselines and implementations, including Simple-RL (Zeng et al., 2025) and Oat-Zero (Liu et al., 2025), are commonly used for evaluation. LUFFY (Yan et al., 2025) explores RLVR with off-policy guidance via prefix-continuation, contrasting standard on-policy GRPO training.

### 2.1. Preliminaries

**GRPO.** GRPO (Shao et al., 2024) is a critic-free policy optimization method commonly used in reinforcement learning from verifiable rewards. Given a prompt $q$, GRPO

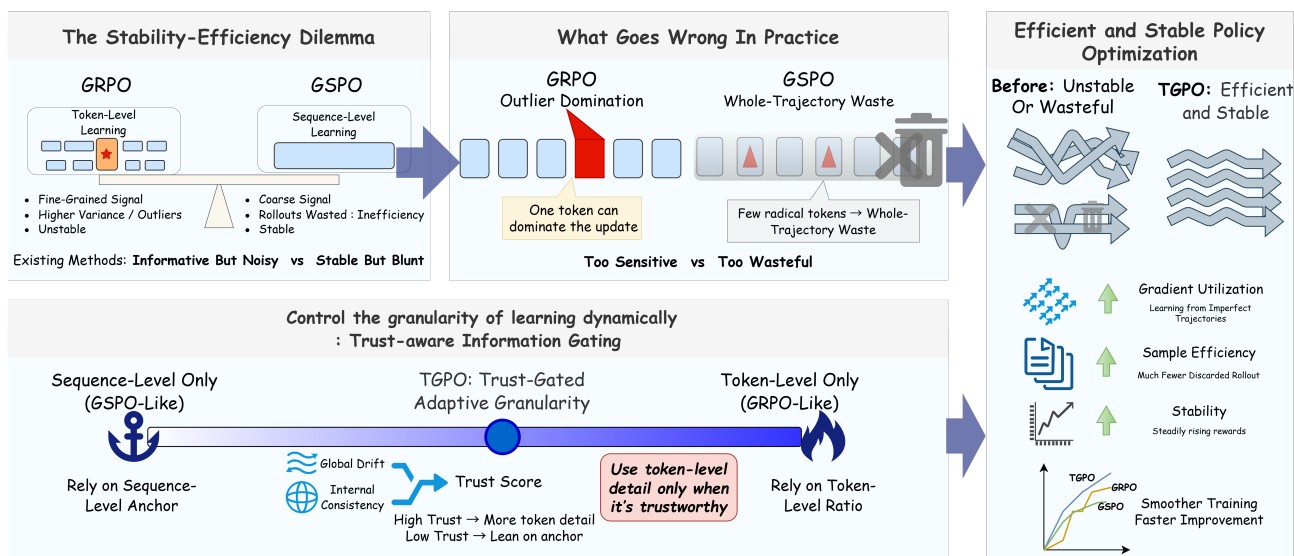

*Figure 1.* **Overview of Trust-Gated Policy Optimization** Policy optimization faces a stability–efficiency tradeoff: token-level updates are expressive but unstable, while sequence-level optimization is stable but wasteful. TGPO resolves this tension by adaptively controlling learning granularity, using token-level detail only when it is trustworthy and otherwise relying on a sequence-level anchor.

samples a group of $G$ rollouts $\{o^i\}_{i=1}^{G}$ from the old policy $\pi_{\theta_{\text{old}}}$ and computes group-normalized advantages:

$$\hat{A}_t^i = \frac{R^i - \text{mean}(\{R^i\}_{i=1}^{G})}{\text{std}(\{R^i\}_{i=1}^{G})}. \qquad (1)$$

The GRPO objective is defined as:

$$\mathcal{J}_{\text{GRPO}}(\theta) = \mathbb{E}_{q\sim\mathcal{D},\,\{o^i\}_{i=1}^{G}\sim\pi_{\theta_{\text{old}}}(\cdot|q)} \left[ \frac{1}{G}\sum_{i=1}^{G}\frac{1}{T_i}\sum_{t=1}^{T_i}\Big( \min \right.$$
$$\left. \big(r_t^i(\theta)\hat{A}_t^i,\ \text{clip}(r_t^i(\theta), 1-\varepsilon, 1+\varepsilon)\hat{A}_t^i\big) - \beta\,\mathbb{D}_{\text{KL}}(\pi_\theta\|\pi_{\text{ref}})\Big) \right]. \qquad (2)$$

Here $r_t^i(\theta) = \frac{\pi_\theta(o_t^i|q,o_{<t}^i)}{\pi_{\theta_{\text{old}}}(o_t^i|q,o_{<t}^i)}$ denotes the token-level importance sampling ratio. Although GRPO supports token-level updates, its optimization process can be noisy. The Token-level IS ratios are estimated from single samples and often exhibit high variance, which accumulates over long sequences. Clipping affects tokens unevenly, which may distort gradient contributions and lead to unstable optimization behavior (Zhao et al., 2025; Zheng et al., 2025). In addition, token-level credit assignment does not explicitly account for sequence-level structure, limiting its ability to guide coherent long-horizon reasoning.

**GSPO.** GSPO (Zheng et al., 2025) improves optimization stability by lifting importance sampling from the token level to the sequence level. Instead of token-wise ratios, GSPO uses a single sequence-level IS ratio for each rollout. The resulting objective is:

$$\mathcal{J}_{\text{GSPO}}(\theta) = \mathbb{E}_{q\sim\mathcal{D},\,\{o^i\}_{i=1}^{G}\sim\pi_{\theta_{\text{old}}}(\cdot|q)} \left[ \frac{1}{G}\sum_{i=1}^{G} \right.$$
$$\left. \min\big(s^i(\theta)\hat{A}_t^i,\ \text{clip}(s^i(\theta), 1-\varepsilon, 1+\varepsilon)\hat{A}_t^i\big) \right]. \qquad (3)$$

Here

$$s^i(\theta) = \left(\frac{\pi_\theta(o^i|q)}{\pi_{\theta_{\text{old}}}(o^i|q)}\right)^{1/|o^i|} = \exp\left(\frac{1}{|o^i|}\sum_{t=1}^{|o^i|}\log r_t^i(\theta)\right), \qquad (4)$$

which corresponds to the geometric mean of token-level log ratios and is applied uniformly to all tokens within the response.

By construction, this formulation reduces sensitivity to individual tokens with extreme likelihood ratios and typically yields more stable updates. However, the sequence-level ratio and advantage are shared equally across all tokens, implicitly assuming homogeneous token-level contribution to the final outcome. This uniform assignment leads to coarse credit allocation, limiting the ability to emphasize positions that are critical for successful reasoning. In practice, maintaining stability further requires conservative clipping ranges, under which a large fraction of otherwise valid rollouts may be discarded, reducing effective gradient utilization and data efficiency.

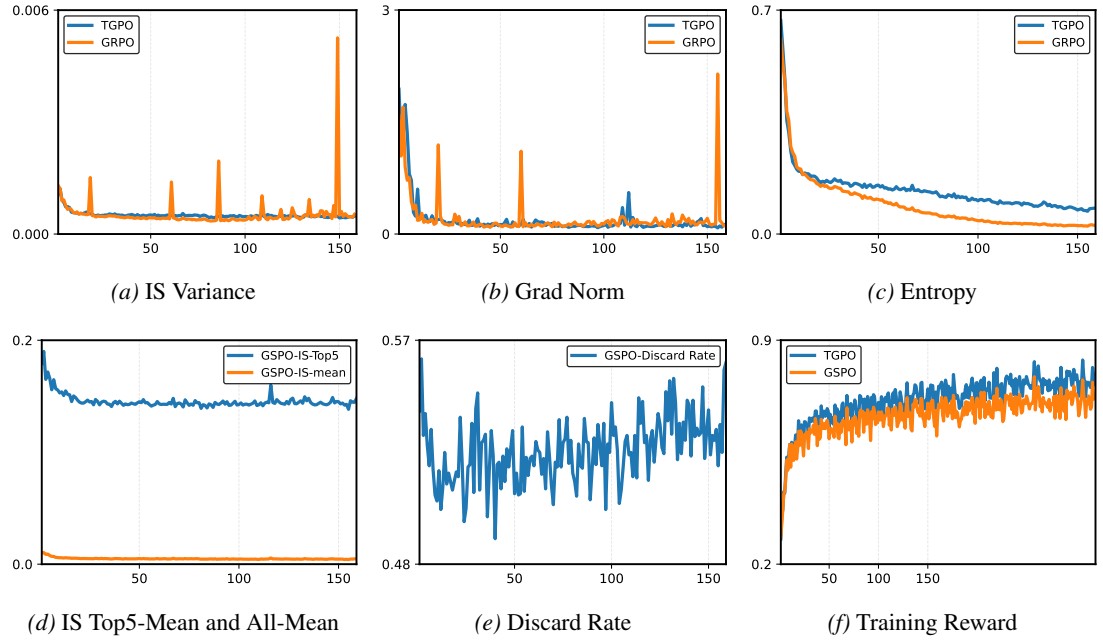

*Figure 2.* Training dynamics of (a–c) TGPO vs. GRPO and (d–f) TGPO vs. GSPO.

## 3. Method

### 3.1. Beyond Token IS vs. Sequence IS

Outcome-supervised RLVR methods such as GRPO provide supervision at the response level, while the optimization objective typically relies on token-wise importance sampling ratios to scale gradients. In this setting, the importance ratio $\rho_t = \frac{\pi_\theta(y_t|x,y_{<t})}{\pi_{\theta_{\text{old}}}(y_t|x,y_{<t})}$ often functions less as a distribution-correction factor and more as a token-level training weight.

Because the advantage signal is largely shared across a response, token-wise weighting can introduce a mismatch. Local likelihood variations that are weakly related to the final outcome may be amplified, resulting in high-variance updates. Recent studies also indicate that, under outcome supervision, naive token-wise importance sampling provides limited practical benefit while exerting a noticeable influence on optimization dynamics (Wang et al., 2025a).

An alternative is to align the optimization granularity with the reward signal. GSPO improves stability by moving importance weighting to the sequence level. This design, however, introduces two inefficiencies. First, the sequence-level ratio remains sensitive to localized deviations, where a small number of outlier tokens can move an entire trajectory beyond the clipping range and suppress otherwise informative gradients. Second, applying a uniform importance weight and advantage to all tokens leads to flat credit assignment, which reduces emphasis on tokens that play a decisive role in reasoning trajectories.

These considerations motivate a trust-aware perspective on importance weighting. Instead of choosing between token-level flexibility and sequence-level robustness, we aim for a mechanism that preserves GSPO-style stability through a sequence-level anchor, while selectively incorporating token-level information when it provides reliable training signals.

### 3.2. Diagnostic Analysis of Token- vs. Sequence-Level Importance Weighting

Figure 2 summarizes diagnostic training dynamics that highlight complementary failure modes of token-level and sequence-level importance weighting under outcome supervision. The message is not that either granularity is universally bad, but that each exhibits a characteristic pathology when used as the sole mechanism to modulate learning.

Panels (a–c) illustrate the instability induced by pure token-level weighting. GRPO frequently produces extreme spikes in IS variance (Fig. 2a), which are accompanied by occasional bursts in gradient norm (Fig. 2b). At the same time, entropy decreases much faster and eventually collapses toward zero (Fig. 2c), indicating that exploration is lost early. Together, these patterns suggest that a small number of outlier tokens can dominate the update, amplifying local likelihood fluctuations into global optimization instability.

Panels (d–f) reveal an orthogonal inefficiency in pure sequence-level weighting. Token-level outliers are still common under GSPO: the top-5 mean ratio is consistently far larger than the sequence-wide mean (Fig. 2d), indicating that a small number of high-IS tokens can create large local devi-

ations. Under sequence-level clipping, such localized deviations are *amplified* into an all-or-nothing decision, pushing the entire response outside the clipping range and discarding otherwise useful trajectories. As a result, GSPO maintains a high discard rate of roughly 45–55% throughout training (Fig. 2e), substantially limiting gradient utilization and data efficiency. Consistently, TGPO achieves faster reward improvement and a better final training reward than GSPO (Fig. 2f), suggesting that overly conservative sequence-level control slows learning by wasting informative responses.

These diagnostics motivate a trust-aware update rule that retains a stable sequence-level reference, while using reliability to adapt the learning granularity (i.e., the strength/weight of fine-grained token information) via an information gate. Next, we formalize this principle as Trust-Gated Policy Optimization (TGPO), with an overview in Fig. 1.

### 3.3. Trust-Gated Policy Optimization

We introduce **Trust-Gated Policy Optimization (TGPO)**, a policy optimization framework that combines sequence-level robustness with adaptive token-level updates. TGPO follows two design principles. Token-level updates are anchored to a robust sequence-level reference to limit the influence of extreme local likelihood fluctuations. Fine-grained token information contributes to learning only when it is reliable, as indicated by both global behavior and internal consistency of the response. From a trust-region perspective, TGPO treats the granularity of policy updates as trajectory-dependent: token-level correction should be used more when the observed policy shift is locally small and coherent, and suppressed when the trajectory exhibits large drift or inconsistent token-level changes.

Given a prompt $x$, a response $y = (y_1, \ldots, y_L)$ is sampled from the behavior policy $\pi_{\theta_{\text{old}}}$. For each token, we compute the log importance ratio

$$\log \rho_t = \log \pi_\theta(y_t \mid x, y_{<t}) - \log \pi_{\theta_{\text{old}}}(y_t \mid x, y_{<t}). \quad (5)$$

Following GSPO, we summarize the response using a sequence-level statistic defined as the geometric mean of token ratios

$$S = \exp\left(\frac{1}{L} \sum_{t=1}^{L} \log \rho_t\right), \quad (6)$$

which provides a stable reference for the overall policy shift on the response.

To evaluate the reliability of token-level information, TGPO computes robust sequence statistics. Because raw $\log \rho_t$ values can be dominated by a small number of extreme tokens, we first apply winsorization within the PPO clipping range:

$$\log \tilde{\rho}_t = \text{clip}(\log \rho_t, \log(1 - \epsilon), \log(1 + \epsilon)). \quad (7)$$

Using the clamped values, we compute the sequence drift

$$\mu = \frac{1}{L} \sum_{t=1}^{L} \log \tilde{\rho}_t, \quad (8)$$

and the within-sequence variance

$$\sigma^2 = \frac{1}{L} \sum_{t=1}^{L} (\log \tilde{\rho}_t - \mu)^2. \quad (9)$$

The drift $\mu$ reflects the global deviation of the response under the new policy, while $\sigma^2$ captures the consistency of token-level likelihood changes. Together, these two quantities provide a second-order summary of the local policy shift, since $\mu^2 + \sigma^2 = \frac{1}{L} \sum_{t=1}^{L} (\log \tilde{\rho}_t)^2$ equals the empirical second moment of the winsorized log-ratios. Thus, the gate is not designed to estimate token importance directly, but to measure whether token-level importance weighting is trustworthy for the current trajectory.

These statistics are mapped to a continuous trust score that regulates the contribution of token-level information. We define

$$G_{\text{drift}} = \exp\left(-\frac{\mu^2}{2\epsilon^2}\right), \qquad G_{\text{cons}} = \exp\left(-\frac{\sigma^2}{2\epsilon^2}\right), \quad (10)$$

and combine them multiplicatively,

$$G = G_{\text{drift}} \cdot G_{\text{cons}} \in (0, 1]. \quad (11)$$

Higher values correspond to responses that are both globally well-behaved and internally consistent. The multiplicative form follows this second-moment interpretation: it is equivalent to an exponential penalty on $\mu^2 + \sigma^2$, normalized by the local update scale $\epsilon$. In this sense, $G$ acts as a soft trust-region proxy, decreasing smoothly when either the global drift or the within-sequence variability becomes large. Gradients are not propagated through $G$, which serves only as a control signal. Based on the trust score, TGPO defines a trust-gated mixing coefficient:

$$\alpha = \frac{1}{\tau} G, \quad (12)$$

where $\tau$ controls the maximum granularity of token-level updates. Token-wise weights are obtained by smoothly combining the sequence anchor and the token-level ratio in log space:

$$\begin{aligned} \log w_t &= (1 - \alpha)\,\mu + \alpha\,\log \rho_t, \\ \Longleftrightarrow \quad w_t &= \tilde{S}^{1-\alpha}\,\rho_t^\alpha, \qquad \tilde{S} = \exp(\mu). \end{aligned} \quad (13)$$

When the trust score is high, TGPO allows greater token-level variation and supports fine-grained credit assignment. When the trust score is low, the update approaches a robust sequence-level weight $w_t \approx \tilde{S}$, recovering a GSPO-style

*Table 1.* Overall performance on seven benchmarks. We compare our method, TGPO, against baselines across diverse architectures, including the Qwen2.5-Math-7B and Llama-3.2-3B-Instruct. Scores: greedy@1 (%). Best results are in **bold**. The background color of TGPO cells indicates performance change vs. other baselines (**green** for improvement, **red** for decline).

| Model | AIME 24 | AIME 25 | AMC | Minerva | MATH-500 | Olympiad | Gaokao2023en | Avg. |
|---|---|---|---|---|---|---|---|---|
| *Qwen2.5-Math-7B* | | | | | | | | |
| Base Model | 13.3 | 13.3 | 42.5 | 16.5 | 53.6 | 18.2 | 35.1 | 27.5 |
| GRPO | 30.0 | 13.3 | 60.0 | 33.4 | 75.8 | 41.3 | 62.6 | 45.2 |
| DAPO | 33.3 | 13.3 | 62.5 | 37.5 | 78.0 | 40.3 | 62.1 | 46.7 |
| GSPO | 26.7 | 6.7 | 60 | 35.7 | 79.6 | 41.5 | 60.3 | 44.4 |
| SimpleRL-Zero | 23.3 | 13.3 | 55.0 | 31.6 | 76.8 | 37.2 | 60.8 | 42.6 |
| Oat-Zero | 30.0 | 16.7 | 62.5 | 34.6 | 78.0 | 41.0 | 62.9 | 46.5 |
| LUFFY | 33.3 | 16.7 | 62.5 | 33.8 | 75.2 | 41.7 | 62.7 | 46.6 |
| **TGPO** | **43.3** | **20.0** | **65.0** | **37.5** | **79.8** | **42.7** | **65.2** | **50.5** |
| *Llama-3.2-3B-Instruct* | | | | | | | | |
| Base Model | 6.7 | 0.0 | 20.0 | 11.8 | 38.3 | 12.6 | 33.5 | 17.6 |
| GRPO | 13.3 | 0.0 | 27.5 | **19.9** | 51.8 | 18.3 | 45.7 | 25.2 |
| DAPO | **20.0** | 0.0 | 22.5 | **19.9** | 54.4 | 19.6 | 47.0 | 26.2 |
| GSPO | 10.0 | 0.0 | 22.5 | 15.8 | 51.6 | 18.5 | 44.4 | 23.3 |
| **TGPO** | 13.3 | **3.3** | **30.0** | 19.5 | **54.6** | **20.3** | **47.8** | **27.0** |

conservative sequence-level update without discarding the response. TGPO incorporates the anchored weights into a PPO-style clipped surrogate objective. For each token, the loss is:

$$\mathcal{L}_t(\theta) = \min\Big(w_t\,\hat{A},\ \mathrm{clip}(w_t, 1 - \epsilon_{low}, 1 + \epsilon_{high})\,\hat{A}\Big),\quad (14)$$

where $\hat{A}$ denotes the group-normalized advantage. The objective is averaged across tokens and responses. Although anchored weighting reduces variance, token-level clipping is retained as a final safeguard.

Overall, TGPO can be viewed as a continuous, trust-conditioned modulation between sequence-level and token-level policy optimization, in which both sequence-level anchors and token-level contributions are adaptively reweighted rather than combined via a fixed mixture. By anchoring updates to robust sequence statistics while selectively admitting token-level information based on its reliability, TGPO alleviates the gradient underutilization inherent in GSPO and reduces the instability associated with naive token-wise importance sampling, thereby enabling stable and sample-efficient optimization under outcome supervision.

## 4. Experiments

### 4.1. Experimental Setup

**Datasets.** Our training data is sourced from the DeepScaleR-Preview-Dataset (Luo et al., 2025), compris-

ing approximately 40k mathematical problems from diverse public benchmarks. Following prior work, we apply model-aware filtering to prioritize problems that are most informative for learning. Using QWEN2.5-MATH-7B (Yang et al., 2024) as a fixed probe, we evaluate each problem with 8 independent samples. Problems consistently solved across all attempts are removed as too easy, while those with partial or no success are retained. This yields a curated dataset that emphasizes both challenge and learnability. All methods in our experiments are trained on this same filtered set.

**Benchmarks and Metrics.** We evaluate our models on a diverse suite of mathematics benchmarks, including GaoKao2023en (Chinese GaoKao Community, 2024), AIME24 (MAA, 2024), AIME25 (MAA, 2025), AMC (MAA, 2023), MATH-500 (Hendrycks et al., 2021), and OlympiadBench (He et al., 2024). Following standard practice, we report `greedy@1`, `avg@32`, and `pass@32`, which correspond to single-sample greedy accuracy, average accuracy across 32 stochastic samples, and the proportion of problems with at least one correct solution among 32 samples, respectively. Unless stated otherwise, stochastic decoding uses the model-recommended settings with temperature 0.5 and top-$p$ 0.8.

**Models and Setups.** We conduct experiments using two model families: Qwen2.5-Math-7B (Yang et al., 2024) and LLaMA-3.2-3B-Instruct (Grattafiori et al., 2024), with a fixed prompt length of 2048 tokens. Due to model-specific

*Table 2.* Overall performance on benchmarks evaluated with Avg@32 and Pass@32. We omit AMC, MATH-500, and Minerva due to space constraints: AMC and MATH-500 are nearly saturated across all models, while Minerva shows minimal variance among methods. Best results are in **bold**. The background color of TGPO cells indicates performance change vs. other baselines (**green** for improvement, **red** for decline).

| Model | AIME 24 | | AIME 25 | | Olympiad | | Gaokao2023en | | AVG | |
|---|---|---|---|---|---|---|---|---|---|---|
| | Avg@32 | Pass@32 | Avg@32 | Pass@32 | Avg@32 | Pass@32 | Avg@32 | Pass@32 | Avg@32 | Pass@32 |
| Qwen2.5-Math-7B | 12.7 | 46.7 | 4.1 | 33.3 | 16.6 | 58.4 | 35.3 | 81.0 | 17.0 | 54.9 |
| GRPO | 28.1 | 46.7 | 5.9 | 30.0 | 40.2 | 57.6 | 60.6 | 75.3 | 33.7 | 52.4 |
| DAPO | **36.5** | 56.7 | 11.6 | 30.0 | 41.5 | 66.5 | 62.0 | 78.7 | 37.9 | 58.0 |
| GSPO | 31.0 | 53.3 | 11.4 | 30.0 | 41.5 | 66.4 | 62.4 | 79.7 | 36.6 | 57.4 |
| **TGPO** | 33.6 | **60.0** | **13.3** | **40.0** | **42.7** | **66.8** | **65.5** | **82.1** | **38.8** | **62.2** |

constraints, the maximum output length is set to 2048 tokens for Qwen2.5 and 8192 tokens for LLaMA. Training is performed for 10 epochs using the VERL framework (Sheng et al., 2025a), with a winsorization clip ratio of 0.2 and a temperature parameter $\tau = 2$. We disable the KL penalty to encourage broader policy exploration, and all remaining hyperparameters follow the default settings from GSPO (Zheng et al., 2025). Reported results are based on the best-performing checkpoint.

**Baselines.** We compare TGPO with three classes of baselines. (1) On-policy RLVR methods, including vanilla GRPO (Shao et al., 2024) and its stronger variants DAPO (Yu et al., 2025) and GSPO (Zheng et al., 2025); (2) optimized GRPO-based implementations such as SimpleRL (Zeng et al., 2025) and Oat-Zero (Liu et al., 2025), which serve as strong public benchmarks; and (3) LUFFY (Yan et al., 2025), which integrates off-policy guidance. All methods are trained on our curated dataset with their recommended hyperparameters. For Simple-RL and Oat-Zero, we report results using publicly released model weights.

### 4.2. Main Results

As shown in Table 1, TGPO consistently outperforms all baselines in Greedy@1 accuracy across most tasks. On Qwen2.5-Math-7B, it achieves the top score on every benchmark, raising the overall average from 46.7 to 50.5—a relative gain of +8.1%. Since Greedy@1 reflects core reasoning under deterministic decoding, this improvement suggests that TGPO enhances fundamental solution quality rather than benefiting from sampling. Similar gains are observed on LLaMA-3.2-3B-Instruct, demonstrating that TGPO generalizes well across model backbones and scales.

Table 2 presents results under sampling-based evaluation, where TGPO again achieves the best overall performance. It improves Avg@32 from 37.9 to 38.8 (+2.4% relative) and Pass@32 from 58.0 to 62.2 (+7.2% relative).

While Avg@32 reflects robustness across diverse rollouts, Pass@32 indicates the achievable upper bound with sufficient exploration. The consistent improvements across Greedy@1, Avg@32, and Pass@32 suggest that TGPO yields well-rounded policy gains—enhancing deterministic reasoning, average-case reliability, and exploratory potential—driven by its stable training dynamics and adaptive information gating.

*Table 3.* Ablation results of TGPO components and $\tau$. Drops are relative to the full TGPO model.

| Model | AIME 24 | AIME 25 | Olympiad | Gaokao2023en | Avg. |
|---|---|---|---|---|---|
| Qwen Base | 13.3 | 13.3 | 18.2 | 35.1 | 20.0 |
| **TGPO** | **43.3** | **20.0** | 42.7 | **65.2** | **42.8** |
| - Anchor | 30.0 | 16.7 | 41.1 | 63.9 | $37.9_{-4.9}$ |
| - Gate | 36.7 | 6.7 | 42.4 | 61.6 | $36.9_{-5.9}$ |
| - Cons | 30.0 | 13.3 | 42.4 | 62.9 | $37.2_{-5.6}$ |
| - Drift | 36.7 | 16.7 | 42.2 | 63.2 | $39.7_{-3.1}$ |
| $\tau = 1$ | 36.7 | 13.3 | 42.2 | 63.2 | $38.9_{-3.9}$ |
| $\tau = 3$ | 36.7 | 13.3 | **42.8** | 63.4 | $39.1_{-3.7}$ |
| GSPO | 26.7 | 6.7 | 41.5 | 60.3 | $33.8_{-9.0}$ |

### 4.3. Ablations of TGPO Components

We conduct ablation studies to assess the contribution of key TGPO components, including the sequence anchor, information gate, trust score formulation, and trust-gating schedule. As summarized in Table 3, each module plays a distinct role in enabling stable and effective policy optimization.

**Effect of TGPO components.** Removing either the sequence anchor or the information gate leads to a clear performance drop, indicating that both are essential for effective optimization. Notably, ablating the information gate causes a larger degradation than removing the anchor alone, highlighting the importance of adaptively regulating token-level contributions. Further ablation of the trust score shows that neither drift nor consistency alone is sufficient, as each

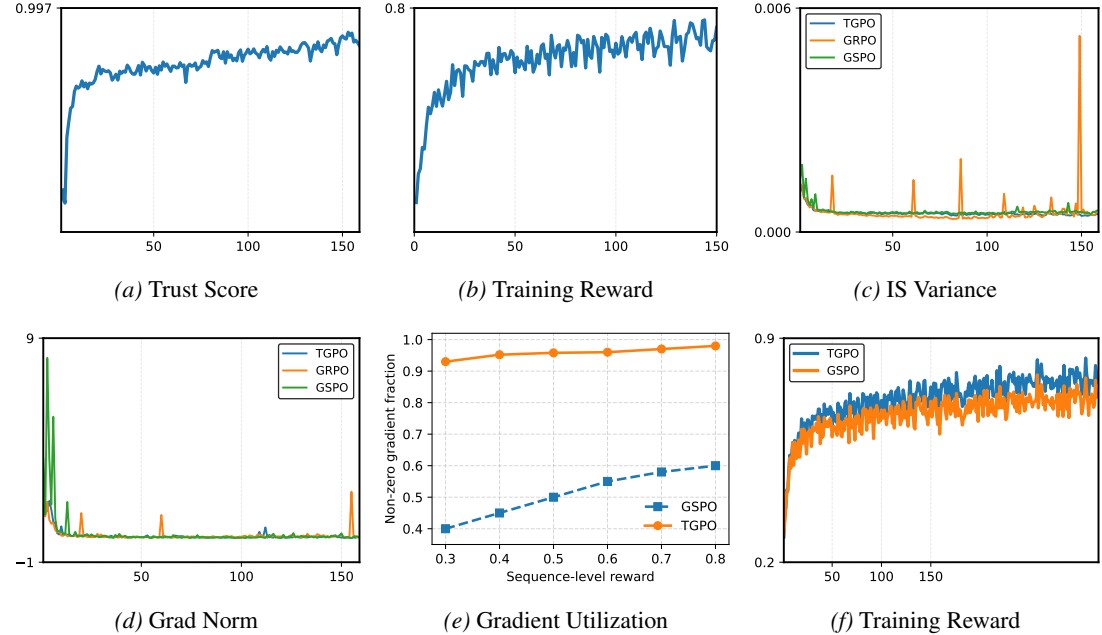

*Figure 3.* Training dynamics related to the Analysis section.

variant underperforms the full design. Among the two, the consistency term has a stronger impact, suggesting that capturing intra-trajectory agreement is particularly important for reliable token-level learning. Overall, these results demonstrate that the sequence anchor and information gate are complementary, and that both drift and consistency are necessary for robust trust estimation.

**Effect of the trust-gating schedule.** We also examine the effect of the trust-gating parameter $\tau$, which governs the granularity of token-level updates. Performance varies only modestly across different values of $\tau$, indicating that TGPO is relatively robust to this hyperparameter. Nevertheless, $\tau = 2$ consistently yields the best overall performance across tasks. This setting corresponds to a trust-modulated geometric reweighting, which reduces to an equal-weight geometric mean of the sequence-level anchor and token-level likelihood ratio when the trust score approaches one, offering a balanced trade-off between optimization stability and fine-grained credit assignment.

## 5. Analysis

We conduct a detailed analysis to examine TGPO's training dynamics and identify the factors contributing to its stability and efficiency, as shown in Figure 3. Unless otherwise specified, the horizontal axis represents training steps, and the vertical axis corresponds to the quantity indicated in each subplot title.

### 5.1. How the Trust Score Behaves

We begin by analyzing the evolution of the trust score during training. As shown in Fig. 3a, the score starts low and steadily increases, closely tracking improvements in training rewards (Fig. 3b). This alignment suggests that trust estimation is well-correlated with learning progress. Early on, low trust suppresses token-level updates, favoring sequence-level signals to stabilize learning when the policy is still unreliable. As training advances and policy consistency improves, rising trust enables finer-grained token-level optimization, accelerating reward gains. This dynamic reflects TGPO's intended behavior—a trust-guided transition from conservative to expressive updates.

### 5.2. Training Stability

We next assess training stability and sample efficiency. As shown in Fig. 3c and Fig. 3d, TGPO exhibits significantly lower importance-sampling variance and more stable gradient norms compared to baselines. In contrast, GSPO displays pronounced early-stage spikes, reflecting unstable optimization. This instability arises from GSPO's sequence-level nullification, which discards partially informative trajectories when the policy is still weak, resulting in noisy and sparse gradient signals. TGPO mitigates this by anchoring updates to sequence-level references while adaptively incorporating token-level information based on trust scores. This leads to more stable early training and faster convergence, as evidenced by the reward curves in Fig. 3f.

## 5.3. "Retain Rather Than Exclude"

We examine how different methods utilize training samples during policy optimization. As shown in Fig. 3e, TGPO achieves a substantially higher fraction of rollouts that contribute non-zero gradients, whereas sequence-level methods often discard entire trajectories once they are deemed imperfect, resulting in low gradient utilization. Instead of excluding such samples, TGPO selectively down-weights unreliable rollouts thus retaining these partially informative sequences, enabling effective gradient extraction across a broader portion of the reward spectrum. This "retain rather than exclude" strategy improves both optimization stability and sample efficiency, leading to faster reward gains as illustrated in Fig. 3f. These findings further explain TGPO's advantage over purely sequence- or token-level optimization approaches.

## 6. Conclusion

We analyze the stability–efficiency trade-off in outcome-supervised policy optimization and attribute it to a mismatch between sequence-level rewards and token-level importance weighting. To address this issue, we propose Trust-Gated Policy Optimization (TGPO), which anchors updates to stable sequence-level references while adaptively controlling token-level learning through trust. This design enables effective use of fine-grained signals from imperfect trajectories without compromising stability or data efficiency. Empirically, TGPO consistently improves both optimization dynamics and final reasoning performance across models and benchmarks. More broadly, our results suggest that learning granularity should be treated as an adaptive, trust-aware property rather than a fixed design choice.

## Impact Statement

This work improves the stability and efficiency of reinforcement learning for large language models trained with outcome-level supervision, without introducing additional data, supervision, or computational cost. By reducing unstable updates and unnecessary rollout discarding, it can make post-training pipelines more reliable and easier to deploy in practice. We do not anticipate direct negative societal impacts; as with all advances in language model training, responsible deployment remains essential.

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

# A. More Information About TGPO

We include additional implementation details of Trust-Gated Policy Optimization (TGPO) in this section.

## A.1. Pesudo Code

This section provides a pseudocode description of Trust-Gated Policy Optimization for reference.

---

**Algorithm 1** Trust-Gated Policy Optimization (TGPO)

---

**Require:** Dataset $\mathcal{D}$; behavior policy $\pi_{\theta_{\text{old}}}$; current policy $\pi_\theta$; Winsorizing clip $\epsilon$; asymmetric clip $(\epsilon_{\text{low}}, \epsilon_{\text{high}})$; trust temperature $\tau$; group size $G$; learning rate $\eta$.

1: **for** each update iteration **do**
2:     Sample a batch of prompts $\{x\} \sim \mathcal{D}$.
3:     **for** each prompt $x$ **do**
4:         Sample a group of responses $\{y^{(i)}\}_{i=1}^G \sim \pi_{\theta_{\text{old}}}(\cdot \mid x)$ with lengths $\{L_i\}$.
5:         Compute group-normalized advantages $\{\hat{A}^{(i)}\}_{i=1}^G$ (shared across tokens within each response).
6:         **for** each response $y^{(i)} = (y_1^{(i)}, \ldots, y_{L_i}^{(i)})$ **do**
7:             **for** $t = 1,\ldots,L_i$ **do**
8:                 $\log \rho_t^{(i)} \leftarrow \log \pi_\theta(y_t^{(i)} \mid x, y_{<t}^{(i)}) - \log \pi_{\theta_{\text{old}}}(y_t^{(i)} \mid x, y_{<t}^{(i)})$
9:                 $\log \tilde{\rho}_t^{(i)} \leftarrow \text{clip}\Big(\log \rho_t^{(i)}, \log(1-\epsilon), \log(1+\epsilon)\Big)$
10:             **end for**
11:             **Sequence anchor:** $S^{(i)} \leftarrow \exp\Big(\frac{1}{L_i} \sum_{t=1}^{L_i} \log \rho_t^{(i)}\Big)$
12:             **Robust stats:** $\mu^{(i)} \leftarrow \frac{1}{L_i} \sum_{t=1}^{L_i} \log \tilde{\rho}_t^{(i)}$; $(\sigma^2)^{(i)} \leftarrow \frac{1}{L_i} \sum_{t=1}^{L_i} \big(\log \tilde{\rho}_t^{(i)} - \mu^{(i)}\big)^2$
13:             **Trust score (stop-grad):** $G^{(i)} \leftarrow \exp\Big(-\frac{(\mu^{(i)})^2}{2\epsilon^2}\Big) \cdot \exp\Big(-\frac{(\sigma^2)^{(i)}}{2\epsilon^2}\Big)$;  **detach**$(G^{(i)})$
14:             **Gate:** $\alpha^{(i)} \leftarrow \frac{1}{\tau} G^{(i)}$
15:             **for** $t = 1,\ldots,L_i$ **do**
16:                 **Anchored token weight:** $\log w_t^{(i)} \leftarrow (1-\alpha^{(i)})\mu^{(i)} + \alpha^{(i)} \log \rho_t^{(i)}$
17:                 $w_t^{(i)} \leftarrow \exp(\log w_t^{(i)}) \{= (S^{(i)})^{1-\alpha^{(i)}}(\rho_t^{(i)})^{\alpha^{(i)}}\}$
18:                 **Clipped surrogate:** $\ell_t^{(i)}(\theta) \leftarrow \min\Big(w_t^{(i)} \hat{A}^{(i)}, \text{clip}(w_t^{(i)}, 1-\epsilon_{\text{low}}, 1+\epsilon_{\text{high}}) \hat{A}^{(i)}\Big)$
19:             **end for**
20:         **end for**
21:     **end for**
22:     $\mathcal{L}(\theta) \leftarrow -\frac{1}{\sum_i L_i} \sum_x \sum_{i=1}^G \sum_{t=1}^{L_i} \ell_t^{(i)}(\theta)$
23:     $\theta \leftarrow \theta - \eta \nabla_\theta \mathcal{L}(\theta)$
24:     $\theta_{\text{old}} \leftarrow \theta$ {on-policy refresh for next iteration}
25: **end for**

---

## A.2. Code

We provide a reference implementation of the TGPO policy loss used in our experiments, illustrating how the trust score, gated mixing coefficient, and anchored weights are computed in practice.

```python
@register_policy_loss("tgpo")
def compute_policy_loss_tgpo(
    old_log_prob: torch.Tensor,
    log_prob: torch.Tensor,
    advantages: torch.Tensor,
    response_mask: torch.Tensor,
    loss_agg_mode: str = "seq-mean-token-mean",
    config: Optional[DictConfig | ActorConfig] = None,
    rollout_is_weights: torch.Tensor | None = None,
) -> tuple[torch.Tensor, torch.Tensor, torch.Tensor, torch.Tensor]:
```

```python
"""
TGPO:_The_Final_Version.

This_ensures_that_we_only_perform_fine-grained_(high_alpha)_updates_when:
1._The_sequence_as_a_whole_is_within_the_trust_region.
2._The_tokens_within_the_sequence_are_consistent_with_each_other.
"""

    assert config is not None
    assert isinstance(config, ActorConfig)

    # 1. Configs
    clip_ratio = config.clip_ratio if config.clip_ratio is not None else 0.2

    default_tau = 2.0
    tau = default_tau
    try:
        if getattr(config, "policy_loss", None) is not None:
            tau = getattr(config.policy_loss, "tgpo_temperature", default_tau)
    except Exception:
        pass
    if tau <= 0: tau = default_tau

    base_alpha = 1.0 / tau

    # 2. Basic Log Ratios (rho)
    log_rho = log_prob – old_log_prob  # (bs, T)

    # 3. Calculate Robust Sequence Stats (Mean & Variance)
    # Clamp inputs first for Robust Statistics (Winsorizing)
    # Using symmetric log bounds for statistical robustness
    robust_log_bound = math.log(1.0 + clip_ratio)
    log_rho_clamped = torch.clamp(log_rho, min=-robust_log_bound, max=robust_log_bound)

    seq_lengths = torch.sum(response_mask, dim=-1).clamp(min=1)

    # Robust Mean (Drift)
    log_s_mean = torch.sum(log_rho_clamped * response_mask, dim=-1) / seq_lengths # (bs,)

    # Robust Variance (Consistency)
    # Var = Mean( (x – Mean)^2 )
    # Note: efficient calculation using the clamped values
    log_rho_centered = (log_rho_clamped – log_s_mean.unsqueeze(-1)) ** 2
    log_s_var = torch.sum(log_rho_centered * response_mask, dim=-1) / seq_lengths # (bs,)

    # 4. Dual-Trust Score Calculation
    # We use clip_ratio as the characteristic scale (sigma) for both drift and deviation.
    sigma_sq = clip_ratio ** 2

    # Drift Trust: Penalize if mean is far from 0
    trust_drift = torch.exp( – (log_s_mean ** 2) / (2 * sigma_sq) )

    # Consistency Trust: Penalize if variance is high
    # We treat sqrt(Var) (StdDev) similarly to Drift.
    # If StdDev > clip_ratio, trust drops significantly.
    trust_consistency = torch.exp( – log_s_var / (2 * sigma_sq) )
```

```
# Combined Trust Score \in (0, 1)
# Detached from computation graph
trust_score = (trust_drift * trust_consistency).detach()

# 5. Dynamic Alpha Calculation
# Low trust -> Low alpha -> Weight converges to Sequence Anchor
dynamic_alpha = (base_alpha * trust_score).unsqueeze(-1) # (bs, 1)

# 6. Calculate Consistency-Aware Anchored Weight
# log(w) = (1 - alpha) * log(S_mean) + alpha * log(rho)
log_s_mean_detached = log_s_mean.detach().unsqueeze(-1)
log_rho_detached = log_rho.detach()

log_w_val = (1.0 - dynamic_alpha) * log_s_mean_detached + dynamic_alpha * log_rho_detached

# Clamp for numerical safety
w_val = torch.exp(torch.clamp(log_w_val, min=-20.0, max=20.0))

# 7. Compute Loss (Standard PPO Surrogate)
ratio = w_val * torch.exp(log_prob - log_prob.detach())

pg_loss_1 = -advantages * ratio
pg_loss_2 = -advantages * torch.clamp(ratio, 1.0 - clip_ratio, 1.0 + clip_ratio)

pg_losses = torch.maximum(pg_loss_1, pg_loss_2)

# 8. Aggregation & Metrics
if rollout_is_weights is not None:
    pg_losses = pg_losses * rollout_is_weights

pg_loss = agg_loss(loss_mat=pg_losses, loss_mask=response_mask, loss_agg_mode=loss_agg_mode)

# Metrics
clipped_mask = (pg_loss_2 > pg_loss_1).float()
pg_clipfrac = verl_F.masked_mean(clipped_mask, response_mask)
ppo_kl = verl_F.masked_mean(-log_rho, response_mask)

return pg_loss, pg_clipfrac, ppo_kl, torch.tensor(0.0, device=pg_loss.device)
```

### A.3. Hyperparameters

This subsection summarizes the key hyperparameters and training configurations used in our TGPO experiments for reproducibility.

## B. Local First-Order Equivalence of TGPO

We provide a local analysis of the trust-gated mixing rule used in TGPO. The goal is to clarify that TGPO does not change the first-order sequence-level policy-gradient direction at the on-policy expansion point. We do not claim that TGPO is globally identical to the standard token-level policy-gradient estimator for all $\theta$. Rather, we show that, at $\theta = \theta_{\text{old}}$, TGPO preserves the same first-order direction as standard REINFORCE / GRPO, and its effect appears only in higher-order terms.

*Table 4.* Key hyperparameters and training configurations for TGPO experiments.

| Category | Hyperparameter | Value |
|---|---|---|
| Algorithm | Advantage estimator | GRPO |
| Algorithm | Loss mode | TGPO |
| Algorithm | TGPO temperature ($\tau$) | 2 |
| Algorithm | Loss aggregation | `seq-mean-token-mean` |
| Algorithm | Token clip (low / high) | 0.20 / 0.28 |
| Algorithm | KL in reward / KL loss | off / off (coef $= 0$) |
| Rollout | Engine / mode | vLLM / sync |
| Rollout | #responses per prompt ($n$) | 8 |
| Rollout | Sampling temperature | 1.0 |
| Rollout | Max prompt / response length | 2048 / 2048 |
| Rollout | GPU memory utilization | 0.8 |
| Rollout | Chunked prefill | enabled |
| Rollout | Max batched tokens | 32768 |
| Optimization | Learning rate | $1 \times 10^{-6}$ |
| Optimization | Weight decay | 0 |
| Optimization | Gradient clip | 1.0 |
| Optimization | Train batch size | 256 |
| Optimization | PPO mini-batch size | 64 |
| Optimization | PPO micro-batch / GPU | 16 |
| Training | Total epochs | 10 |
| Training | Validation batch size | 512 |
| Training | Eval / save frequency (epochs) | 10 / 20 |
| System | GPUs per node / #nodes | 8 / 1 |
| System | Tensor parallel size (rollout) | 2 |
| System | Gradient checkpointing | enabled |

Consider a response $y = (y_1, \ldots, y_L)$ sampled from the behavior policy $\pi_{\theta_{\text{old}}}$. In outcome-supervised RLVR, the advantage $\hat{A}$ is a sequence-level scalar shared across tokens. TGPO defines the token-level log-ratio as

$$\log \rho_t(\theta) = \log \pi_\theta(y_t \mid x, y_{<t}) - \log \pi_{\theta_{\text{old}}}(y_t \mid x, y_{<t}). \tag{15}$$

The robust sequence drift is

$$\mu(\theta) = \frac{1}{L} \sum_{k=1}^{L} \log \tilde{\rho}_k(\theta), \tag{16}$$

where $\log \tilde{\rho}_k$ denotes the winsorized log-ratio. The anchored token weight is defined as

$$\log w_t(\theta) = (1 - \alpha)\mu(\theta) + \alpha \log \rho_t(\theta), \tag{17}$$

or equivalently,

$$w_t(\theta) = \exp\left((1 - \alpha)\mu(\theta) + \alpha \log \rho_t(\theta)\right). \tag{18}$$

Here $\alpha = G/\tau$ is computed from the trust score and treated as a stop-gradient control signal. Therefore, in the following local differentiation, $\alpha$ is fixed with respect to $\theta$.

We analyze the local behavior at the on-policy point $\theta = \theta_{\text{old}}$. At this point, for every token $t$,

$$\rho_t(\theta_{\text{old}}) = 1, \qquad \log \rho_t(\theta_{\text{old}}) = 0. \tag{19}$$

Since the winsorization interval contains zero, winsorization is inactive in a neighborhood of $\theta_{\text{old}}$. Hence,

$$\log \tilde{\rho}_t(\theta_{\text{old}}) = \log \rho_t(\theta_{\text{old}}) = 0, \tag{20}$$

and

$$\mu(\theta_{\text{old}}) = \frac{1}{L} \sum_{k=1}^{L} \log \tilde{\rho}_k(\theta_{\text{old}}) = 0. \tag{21}$$

It follows that

$$\log w_t(\theta_{\mathrm{old}}) = (1 - \alpha) \cdot 0 + \alpha \cdot 0 = 0, \tag{22}$$

and therefore

$$w_t(\theta_{\mathrm{old}}) = 1. \tag{23}$$

Next, because winsorization is locally inactive around zero, its derivative matches that of the unclipped log-ratio. Thus,

$$\nabla_\theta \mu(\theta)\big|_{\theta=\theta_{\mathrm{old}}} = \frac{1}{L} \sum_{k=1}^{L} \nabla_\theta \log \rho_k(\theta)\big|_{\theta=\theta_{\mathrm{old}}}. \tag{24}$$

Since $\alpha$ is stop-gradient, differentiating $\log w_t$ gives

$$\nabla_\theta \log w_t(\theta)\big|_{\theta=\theta_{\mathrm{old}}} = (1-\alpha)\nabla_\theta \mu(\theta)\big|_{\theta=\theta_{\mathrm{old}}} + \alpha\nabla_\theta \log \rho_t(\theta)\big|_{\theta=\theta_{\mathrm{old}}}. \tag{25}$$

Substituting the expression for $\nabla_\theta \mu$ yields

$$\nabla_\theta \log w_t(\theta)\big|_{\theta=\theta_{\mathrm{old}}} = (1-\alpha)\frac{1}{L}\sum_{k=1}^{L} \nabla_\theta \log \rho_k(\theta)\big|_{\theta=\theta_{\mathrm{old}}} + \alpha\nabla_\theta \log \rho_t(\theta)\big|_{\theta=\theta_{\mathrm{old}}}. \tag{26}$$

Because $w_t(\theta_{\mathrm{old}}) = 1$, we have

$$\nabla_\theta w_t(\theta)\big|_{\theta=\theta_{\mathrm{old}}} = w_t(\theta_{\mathrm{old}})\nabla_\theta \log w_t(\theta)\big|_{\theta=\theta_{\mathrm{old}}} = \nabla_\theta \log w_t(\theta)\big|_{\theta=\theta_{\mathrm{old}}}. \tag{27}$$

Therefore,

$$\nabla_\theta w_t(\theta)\big|_{\theta=\theta_{\mathrm{old}}} = (1-\alpha)\frac{1}{L}\sum_{k=1}^{L} \nabla_\theta \log \rho_k(\theta)\big|_{\theta=\theta_{\mathrm{old}}} + \alpha\nabla_\theta \log \rho_t(\theta)\big|_{\theta=\theta_{\mathrm{old}}}. \tag{28}$$

Now summing over all tokens in the response gives

$$\sum_{t=1}^{L} \nabla_\theta w_t(\theta)\big|_{\theta=\theta_{\mathrm{old}}} = \sum_{t=1}^{L} \left[ (1-\alpha)\frac{1}{L}\sum_{k=1}^{L} \nabla_\theta \log \rho_k(\theta)\big|_{\theta=\theta_{\mathrm{old}}} + \alpha\nabla_\theta \log \rho_t(\theta)\big|_{\theta=\theta_{\mathrm{old}}} \right] \tag{29}$$

$$= (1-\alpha)\sum_{k=1}^{L} \nabla_\theta \log \rho_k(\theta)\big|_{\theta=\theta_{\mathrm{old}}} + \alpha\sum_{t=1}^{L} \nabla_\theta \log \rho_t(\theta)\big|_{\theta=\theta_{\mathrm{old}}} \tag{30}$$

$$= \sum_{t=1}^{L} \nabla_\theta \log \rho_t(\theta)\big|_{\theta=\theta_{\mathrm{old}}}. \tag{31}$$

Since $\pi_{\theta_{\mathrm{old}}}$ is fixed, we have

$$\nabla_\theta \log \rho_t(\theta) = \nabla_\theta \log \pi_\theta(y_t \mid x, y_{<t}). \tag{32}$$

Therefore,

$$\sum_{t=1}^{L} \nabla_\theta w_t(\theta)\big|_{\theta=\theta_{\mathrm{old}}} = \sum_{t=1}^{L} \nabla_\theta \log \pi_\theta(y_t \mid x, y_{<t})\big|_{\theta=\theta_{\mathrm{old}}}. \tag{33}$$

Multiplying by the shared sequence-level advantage $\hat{A}$, the first-order TGPO update recovers the standard sequence-level policy-gradient direction:

$$\hat{A}\sum_{t=1}^{L} \nabla_\theta w_t(\theta)\big|_{\theta=\theta_{\mathrm{old}}} = \hat{A}\sum_{t=1}^{L} \nabla_\theta \log \pi_\theta(y_t \mid x, y_{<t})\big|_{\theta=\theta_{\mathrm{old}}}. \tag{34}$$

Up to the common normalization by sequence length used in implementation, this is exactly the REINFORCE / GRPO sequence-level policy-gradient direction.

This result shows that TGPO does not introduce first-order bias at the behavior policy. The trust-gated mixing changes only the higher-order behavior of the local surrogate: when token-level ratios are reliable, the update remains closer to token-wise correction; when they exhibit large drift or high within-sequence inconsistency, the update falls back toward a lower-variance sequence-level anchor. In this sense, the gate acts as a controlled higher-order regularizer rather than an uncontrolled distortion of the policy-gradient direction.

## C. Future Directions

TGPO introduces trust-gated control of optimization granularity under outcome supervision. A long-term direction is to investigate whether trust-aware mechanisms can be integrated into more general learning systems, where the notion of trust emerges from richer model-internal or environment-level signals. More broadly, we view trust-gated control as a general design principle for balancing stability and expressiveness in policy optimization, whose implications may extend beyond the specific formulation studied in this work. Exploring these directions would require advances beyond the current experimental scope and is left for future work.

