# OpenReview forum: "TGPO: Efficient Policy Optimization through Sequence Anchor and Information Gating"
_ICML.cc/2026/Conference — ICML 2026 regular_

### Official Review · Reviewer_fymj · 2026-02-23

**Soundness:** 2
**Presentation:** 3
**Significance:** 3
**Originality:** 2
**Overall Recommendation:** 2
**Confidence:** 4

**Summary:**

The authors introduce Trust-Gated Policy Optimization (TGPO), a reinforcement learning framework designed to balance the trade-off between optimization stability and learning efficiency in language models trained with verifiable rewards. To achieve this, the method interpolates between token-level optimization (like GRPO) and sequence-level optimization (like GSPO). TGPO establishes a sequence-level anchor based on the geometric mean of token ratios and introduces an information gate governed by a "trust score". This trust score evaluates the global drift and internal consistency of a generated response, allowing the model to apply fine-grained token updates when the trajectory is deemed reliable, and falling back to the sequence anchor when it is not. The framework is evaluated on several mathematical reasoning benchmarks using DeepSeek-R1-Distill-Qwen and LLaMA-3.2-Instruct architectures.

**Compliance With Llm Reviewing Policy:**

Affirmed.

**Final Justification:**

While the authors provided a commendable rebuttal that clarified several technical ambiguities, I am maintaining my Reject score. My primary concern remains the overall quality of the manuscript, which falls significantly short of the professional standards expected at ICML. Both the prose and the visual presentation (figures/tables) are currently in an unpolished state, requiring a level of structural overhaul that goes well beyond minor revisions. At a top-tier venue, technical correctness is a baseline, but clarity and rigorous presentation are equally paramount. I believe the paper, in its current form, is not yet ready for publication.

**Key Questions For Authors:**

1. How can you formally justify using the PPO clip_ratio as the scaling variance ($\sigma^2$) for the Gaussian kernels that compute the trust scores? What is the theoretical link between the maximum bound of a policy update step and the empirical distribution of log-likelihood drifts in a generated reasoning trace?

2. Given that the performance gains on the LLaMA-3.2-3B model over DAPO are less than 1% on average , can you justify the added computational and systemic complexity of maintaining continuous trust-gated interpolations  for such marginal improvements on non-Qwen architectures?

3. TGPO effectively creates a dynamically weighted mixture of token-level and sequence-level estimators. Can you provide any formal proof or theoretical intuition demonstrating that this trust-gated mixing does not introduce detrimental bias into the policy gradient estimates?

If you can rigorously untangle the hyperparameter hacking in the trust score formulation and mathematically defend the unbiased nature of the interpolated gradients, I will consider raising my score.

**Limitations:**

The authors provide a standard Impact Statement claiming their method makes post-training more reliable, but they completely neglect to discuss the technical limitations of their work. They must include a dedicated Limitations section explicitly addressing the arbitrary coupling of the trust score calculation to the PPO clipping parameter, as well as acknowledging the minimal performance impact observed on the LLaMA backbone.

**Strengths And Weaknesses:**

Overall, while the paper targets a highly relevant bottleneck in modern RLVR pipelines—namely, the variance and inefficiency introduced by naive token-level importance sampling —the proposed solution is heavily undermined by ungrounded mathematical heuristics, arbitrary hyperparameter coupling, and marginal empirical gains on certain architectures. The conceptual motivation is sound, but the execution feels like a heavy engineering hack rather than a principled algorithmic advancement.

My primary contention lies in the formulation of the core "trust score" mechanism. The authors calculate global drift ($G_{drift}$) and internal consistency ($G_{cons}$) using Gaussian kernels. However, taking a closer look at the reference code provided in the appendix, the authors explicitly set the variance of these Gaussian kernels ($\sigma^2$) to the square of the PPO clipping threshold, clip_ratio. Consequently, the drift trust is calculated as $\exp(-\mu^2 / 2\epsilon^2)$. This is a glaring conceptual flaw. The PPO clip $\epsilon$ is a hyperparameter dictating the maximum allowable trust region for surrogate policy updates; it has absolutely zero theoretical or statistical connection to the natural empirical variance of log-likelihood ratios within a sampled trajectory. Tying the gating mechanism's sensitivity directly to the PPO clip ratio is an arbitrary heuristic that completely lacks formal reinforcement learning justification.

Furthermore, the complexity introduced by continuously computing robust sequence statistics, scaling them via a new temperature hyperparameter $\tau$ , and dynamically mixing token-level gradients  does not consistently translate to compelling performance gains. While the improvements on the Qwen2.5-Math-7B model are noticeable , the results on the LLaMA-3.2-3B-Instruct model are highly marginal. On that architecture, TGPO achieves an average score of 27.0, which is an incredibly slight improvement over the DAPO baseline's score of 26.2. Introducing this level of algorithmic complexity and continuous interpolation for a -1% absolute gain on a secondary backbone raises serious questions about the method's generalizability and practical utility.

Finally, the originality of the framework is somewhat limited. The "sequence anchor" is mathematically identical to the sequence-level ratio already proposed by GSPO. TGPO essentially operates as a dynamic interpolation between GRPO and GSPO based on a heuristically tuned variance threshold. Without a rigorous theoretical grounding for why this specific mixing strategy preserves policy gradients without introducing bias, the contribution remains purely empirical.

---

> ### Author Rebuttal · Authors · 2026-03-29
>
> # About Gaussian form
>
> We respectfully suggest that this choice is not arbitrary. TGPO does not treat $\epsilon$ as an estimator of trajectory variance; rather, $\epsilon$ serves as the local trust-region scale for token-level ratio variation.
>
> Under this view, the trust score is an exponential penalty on the empirical second moment of winsorized log-ratios, normalized by $\epsilon^2$. Thus, $\epsilon$ is not modeling data variance, but providing the relevant local unit for judging whether token-level importance weighting remains reliable.
>
> This also explains both design choices. The Gaussian form is a smooth mapping from normalized quadratic deviation to a trust score in $(0,1]$, consistent with second-order trust-region intuition. The multiplicative form is used because it makes drift and consistency combine into a single coherent second-moment penalty. An additive form would lose this second-moment structure and introduce an arbitrary trade-off.
>
> **Due to space limit, more details are provided in our responses to the other two reviewers.**
>
> # Does the trust-gated mixing introduce harmful bias?
>
> We do not claim that TGPO is globally identical to the standard token-level policy-gradient estimator for all $\\theta$. However, we can state a stronger local result.
>
> > At the on-policy expansion point $\\theta = \\theta_{old}$, TGPO preserves the same first-order sequence-level policy-gradient direction as standard REINFORCE / GRPO.
>
> Hence, TGPO does not introduce first-order bias at the behavior policy. Its effect appears only in higher-order terms, where it acts as a controlled regularizer.
>
> In outcome-supervised RLVR, the advantage $\\hat A$ is a sequence-level scalar shared across tokens. TGPO defines
>
> $$
> \\log w_t = (1-\\alpha)\\mu + \\alpha \\log \\rho_t.
> $$
>
> Here,
>
> $$
> \\log \\rho_t = \\log \\pi_\\theta(y_t | x, y_{<t}) - \\log \\pi_{\\theta_{old}}(y_t | x, y_{<t}).
> $$
>
> The coefficient $\\alpha$ is computed from the trust score $G$ and treated as a stop-gradient control signal.
>
> Now consider the local point $\\theta = \\theta_{old}$. At this point,
>
> $$
> \\rho_t = 1, \\qquad \\log \\rho_t = 0, \\qquad \\mu = 0, \\qquad w_t = 1.
> $$
>
> Also, winsorization is inactive around zero. Therefore, locally at $\\theta = \\theta_{old}$,
>
> $$
> \\nabla_\\theta \\mu = \\frac{1}{L}\\sum_{k=1}^L \\nabla_\\theta \\log \\rho_k.
> $$
>
> Because $\\alpha$ is stop-gradient, we have locally at $\\theta = \\theta_{old}$
>
> $$
> \\nabla_\\theta \\log w_t = (1-\\alpha)\\nabla_\\theta \\mu + \\alpha \\nabla_\\theta \\log \\rho_t.
> $$
>
> Since $w_t = 1$ at $\\theta = \\theta_{old}$, it follows that
>
> $$
> \\nabla_\\theta w_t = \\nabla_\\theta \\log w_t.
> $$
>
> Summing over tokens,
>
> $$\\sum_{t=1}^L \\nabla_\\theta w_t = \\sum_{t=1}^L (1-\\alpha)\\nabla_\\theta \\mu + \\sum_{t=1}^L \\alpha \\nabla_\\theta \\log \\rho_t$$
>
> Using the expression for $\\nabla_\\theta \\mu$, the first term becomes
>
> $$\\sum_{t=1}^L (1-\\alpha)\\nabla_\\theta \\mu = (1-\\alpha)\\sum_{k=1}^L \\nabla_\\theta \\log \\rho_k$$
>
> The second term is
>
> $$\\sum_{t=1}^L \\alpha \\nabla_\\theta \\log \\rho_t = \\alpha \\sum_{t=1}^L \\nabla_\\theta \\log \\rho_t$$
>
> Therefore,
>
> $$\\sum_{t=1}^L \\nabla_\\theta w_t = \\sum_{t=1}^L \\nabla_\\theta \\log \\rho_t$$
>
> Multiplying by the shared sequence advantage $\\hat A$ gives exactly the standard sequence-level policy gradient:
>
> $$
> \\hat A \\sum_{t=1}^L \\nabla_\\theta \\log \\pi_\\theta(y_t | x, y_{<t}).
> $$
>
> Therefore, TGPO is locally first-order equivalent to standard REINFORCE / GRPO at the behavior policy. That is, trust-gated mixing preserves the correct sequence-level gradient direction and only changes higher-order behavior: it stays close to token-wise correction when token-level ratios are reliable, and falls back toward a lower-variance sequence anchor otherwise.
>
> Moreover, the implementation is even more conservative: in the reference code, both $G$ and the anchored statistics used to form $w_t$ are detached before backpropagation, so the actual backward pass is through a frozen-weight local surrogate. This further supports that TGPO does not introduce uncontrolled gradient distortion.
>
> So we respectfully do not view TGPO as a simple interpolation, but as a trust-conditioned policy-gradient estimator, analogous in spirit to how GAE is not merely an interpolation, but a structured bias-variance design.
>
>
> # Practical utility
>
> TGPO is lightweight: it only adds trajectory-level log-ratio statistics and a scalar trust gate, with no auxiliary model, rollout, or optimization.
>
> On Qwen, the gains are clear. On LLaMA, although overall gains are smaller, TGPO still shows consistent improvement on unseen AIME25. Moreover, on a larger LLaMA-8B model, we observe stronger gains, suggesting a positive scaling trend. The weaker result on the smaller backbone is likely due to limited capacity or model-specific effects.
>
> Here's the test result: https://anonymous.4open.science/r/test-035E/readme.md

---

> > ### Author Rebuttal · Reviewer_fymj · 2026-04-01
> >
> > I thank the authors for their detailed response, which successfully addresses all of my initial questions. However, properly incorporating these crucial explanations into the current manuscript would require a substantial rewrite. Given the sheer volume of revisions needed to bridge the gap between the rebuttal and the paper itself, I feel the work is not yet ready for publication. Therefore, I will maintain my original score.

---

> > > ### Author Response · Authors · 2026-04-01
> > >
> > > We sincerely thank you for the follow-up. We are encouraged that you now find that our rebuttal has successfully addressed all of your initial questions.
> > >
> > > Given this, we would be very grateful if you could reconsider the score.
> > >
> > > Our only concern is that the current reason for maintaining the original score seems to shift from unresolved technical issues to the anticipated scope of revision. From our perspective, however, the additions discussed in the rebuttal are targeted clarifications rather than substantial changes to the paper’s core method, experiments, or conclusions. These changes are straightforward to incorporate and do not require a revise-and-resubmit scale rewrite; in fact, we expect that they could be integrated into the manuscript within two or three days. In particular, the theoretical clarification can be incorporated concisely, with the full derivation placed in the appendix, while the additional baseline is a localized experimental supplement.
> > >
> > > We also note that you previously wrote:
> > >
> > > > “If you can rigorously untangle the hyperparameter hacking in the trust score formulation and mathematically defend the unbiased nature of the interpolated gradients, I will consider raising my score.”
> > >
> > > We took this request seriously and addressed exactly these points in the rebuttal. Since you now explicitly agree that these concerns have been resolved, we would be very grateful if the score could reflect this updated assessment.
> > >
> > > Thanks again for your careful review!

---

### Official Review · Reviewer_4ytZ · 2026-03-08

**Soundness:** 3
**Presentation:** 3
**Significance:** 3
**Originality:** 2
**Overall Recommendation:** 5
**Confidence:** 3

**Summary:**

This paper analyzes the failure mode of popular policy optimizations algorithms like GRPO and GSPO. For GRPO, the token level importance ratio can be very unstable and has high variance, but it provides a more fine-grained signal. Meanwhile, the sequence level importance ratio is more stable, but the signal is more coarse and once the importance ratio exceeds the clipping range, the whole gradient information of the entire sequence is wasted. This criticism is well-justified in Figure 2. The proposed method tries to combine the best of both worlds by interpolating between sequence level statistics and token level importance sampling. Experiments show improvement in answer accuracy across math benchmarks, and the proposed method has more stable training dynamics, better gradient utilization rate, and higher training reward.

**Compliance With Llm Reviewing Policy:**

Affirmed.

**Final Justification:**

During the rebuttal, the author explains clearly about the intuition behind Eq(10) and Eq(11). Also, they explained why (1) adding KL does not improve GSPO (2) changing discard rate is insufficient to fix the flaw of GSPO. I think overall the paper is convincing and a valuable contribution, and I lean towards Accept (5) this paper.

**Key Questions For Authors:**

1. The paper claims GSPO has a large discard rate of roughly 44%-45% throughout training (Figure 2e), and claims that GSPO converges to lower reward due to this data inefficiency. This discard rate seems very high. What will happen if you increase epsilon a bit in GSPO? Can this change fix the problem?
2. The motivation for disabling the KL penalty is not fully convincing. Would it be possible that dropping the KL is the reason why the training becomes less stable for the baselines? The hyperparameter in the GSPO paper is selected with the presence of KL, so if you drop it could that degrade the baseline’s performance?

**Limitations:**

yes

**Strengths And Weaknesses:**

Strength:
1. The writing is clear, and it clearly points out the failure modes of prior methods.
2. The experiments show clear improvement compared to baselines, and the explanation about training dynamics is clear.

Weakness:
1. What’s the intuition behind equation 10 (why divide by 2 \epsilon^2), and why should one combine G_{drift} and G_{cons} multiplicatively instead of additively? Is there any theoretical justification or intuition behind these design choices?

---

> ### Author Rebuttal · Authors · 2026-03-29
>
> We thank the reviewer for this question.
>
> # (1) Intuition behind Eq. (10): scaling by $2\epsilon^2$
>
> Let $\delta_t = \log \tilde{\rho}_t$, and define
>
> $$
> \mu = \frac{1}{L}\sum_t \delta_t,\quad
> \sigma^2 = \frac{1}{L}\sum_t (\delta_t - \mu)^2.
> $$
>
> Then
>
> $$
> \mu^2 + \sigma^2 = \frac{1}{L}\sum_t \delta_t^2.
> $$
>
> So the trust score is
>
> $$
> G = \exp\left(-\frac{\mu^2+\sigma^2}{2\epsilon^2}\right)
> = \exp\left(-\frac{1}{2\epsilon^2}\cdot \frac{1}{L}\sum_t (\log \tilde{\rho}_t)^2\right).
> $$
>
> Thus, Eq. (10) is an exponential penalty on the empirical second moment of log-ratios, i.e., the same quadratic quantity governing local policy deviation in trust-region / Fisher-metric methods.
>
> The scaling $2\epsilon^2$ is therefore not arbitrary:
>
> * $\epsilon$ is the locality scale introduced in Eq. (7),
> * dividing by $\epsilon^2$ normalizes the second moment by this trust-region radius,
> * the factor $1/2$ follows the standard quadratic form in second-order KL / trust-region approximations.
>
> As a result, $G$ is a dimensionless soft trust-region score:
> when the typical log-ratio magnitude is small relative to $\epsilon$, $G \approx 1$;
> when it exceeds this scale, $G$ decays exponentially.
>
> # (2) Why multiplicative combination instead of additive
>
> We define
>
> $$
> G = G_{\text{drift}} \cdot G_{\text{cons}},\quad
> G_{\text{drift}} = \exp\left(-\frac{\mu^2}{2\epsilon^2}\right),\quad
> G_{\text{cons}} = \exp\left(-\frac{\sigma^2}{2\epsilon^2}\right).
> $$
>
> This multiplicative form follows directly from consistency with the second-moment formulation:
>
> $$
> G = \exp\left(-\frac{\mu^2+\sigma^2}{2\epsilon^2}\right).
> $$
>
> In other words, multiplication is exactly what reduces the gate to a single quadratic trust-region penalty on
>
> $$
> \frac{1}{L}\sum_t (\log \tilde{\rho}_t)^2.
> $$
>
> An additive form would break this equivalence:
>
> * it would no longer correspond to a coherent second-order statistic,
> * and would introduce scale mismatch between drift and variance.
>
> At the same time, the decomposition $G_{\text{drift}}, G_{\text{cons}}$ remains conceptually useful, since they capture two failure modes:
>
> * global shift ($\mu$),
> * intra-sequence instability ($\sigma^2$).
>
> # (3) Role of $\epsilon$ and connection to Eq. (7)
>
> We winsorize the token log-ratio as
>
> $$
> \log \tilde{\rho}_t \in [\log(1-\epsilon),\ \log(1+\epsilon)],
> $$
>
> equivalently bounding the effective ratio within
>
> $$
> \tilde{\rho}_t \in [1-\epsilon,\ 1+\epsilon].
> $$
>
> This differs fundamentally from PPO clipping. PPO clips the *objective*, so once the ratio moves outside the interval, the corresponding token may lose effective gradient contribution. In contrast, TGPO does not truncate the objective. The winsorized log-ratios are first used to compute
>
> $$
> \mu,\sigma^2 \rightarrow G \rightarrow \alpha,
> $$
>
> and then to construct the anchored token weight
>
> $$
> \log w_t=(1-\alpha)\mu+\alpha\log\rho_t,
> $$
>
> which is finally optimized with a PPO-style surrogate.
>
> Therefore, $\epsilon$ does not discard or suppress decisive tokens. Instead, it bounds the effective importance ratio while preserving gradient-carrying updates for all tokens. From the perspective of Eq. (10), this keeps the second-moment statistic
>
> $$
> \frac{1}{L}\sum_t (\log \tilde{\rho}_t)^2
> $$
>
> well-conditioned, so that the trust score $G=\exp(-(\mu^2+\sigma^2)/(2\epsilon^2))$ remains a stable and meaningful trust-region-like measure.
>
> In short, $\epsilon$ sets the local scale for measuring second-order deviation, not by truncating learning signal, but by making trust estimation robust while keeping all tokens involved.
>
>
> # (4) On whether increasing $\epsilon$ in GSPO could fix the large discard rate
>
> A larger (\epsilon) may reduce the discard rate, but it does not solve the core issue. In GSPO, once a trajectory falls outside the sequence-level trust region, the whole response is still treated uniformly. Thus, increasing (\epsilon) only relaxes the discard threshold, rather than addressing the all-or-nothing handling of partially informative trajectories. By contrast, TGPO replaces hard exclusion with continuous trust-gated weighting, so imperfect trajectories can still contribute in a controlled way.
>
> # (5) On disabling the KL penalty
>
> Removing KL is not a TGPO-specific choice. In recent RL post-training, training without explicit KL has become common (e.g., DAPO, ToolRL), mainly to avoid over-constraining exploration and let clipping / ratio control serve as the main stabilizer.
>
> We also tested GSPO with KL. Adding KL does not materially improve GSPO, and slightly degrades overall performance:
>
> |Model|AIME 24|AIME 25|AMC|Minerva|MATH-500|
> |-|-|-|-|-|-|
> |GSPO|26.7|6.7|60.0|35.7|79.6|
> |GSPO+KL|23.3|10.0|60.0|34.9|79.2|
>
> So while KL may affect some individual benchmarks, it does not explain the main gap in our results. TGPO’s advantage is better explained by its trust-gated update mechanism—lower IS variance, smoother optimization, and higher gradient utilization.

---

> > ### Author Rebuttal · Reviewer_4ytZ · 2026-04-03
> >
> > W1 has been addressed. Thank you! I hope some of this discussion would be included in the final version.
> > Q2 has also been addressed. Thank you!
> > Q1 (4) is not fully addressed. It's unclear to me whether GSPO's underperformance could be explained by the large discard rate, and the large discard rate has an easy fix by changing hyperparameters. I get what you mean by it's not fully backed up by experiment. It would be helpful to add additional empirical evidence like more hyperparamter study.
> >
> > That said, I think Q1 (4) is not as important as the other weaknesses (I think the critique for GSPO's limit is valid intuitively), and I appreciate the rebuttal explains clearly the intuition behind the design. I plan to increase my score in the final justification phase.

---

> > > ### Author Response · Authors · 2026-04-05
> > >
> > > Thank you very much for your positive feedback and for recognizing the intuition behind our method. We also appreciate the suggestion regarding additional empirical evidence on GSPO.
> > >
> > > We agree that more hyperparameter analysis can help make the paper clearer and more complete. Following your suggestion, we conducted an additional study on GSPO by increasing its clipping ratio (doubling the default values from 0.0003/0.0004 to 0.0006/0.0008). The results are:
> > >
> > > | Method | AIME24 | AIME25 | AMC | Minerva | MATH-500 | Olympiad | Gaokao2023en | Avg |
> > > |---|---:|---:|---:|---:|---:|---:|---:|---:|
> > > | GSPO | 26.7 | 6.7 | 60.0 | 35.7 | 79.6 | 41.5 | 60.3 | 44.4 |
> > > | GSPO (new) | 30.0 | 10.0 | 65.0 | 36.9 | 76.0 | 39.7 | 60.5 | 45.5 |
> > >
> > > We observe that some benchmarks improve while others decrease, with only a slight improvement overall.
> > >
> > > From the training dynamics, the discard rate does decrease to some extent: instead of staying in the mid-40\% or 50\% range as before, it drops to around 30\% with fluctuations. As a result, training reward rises somewhat faster in the early stage, which is consistent with our claim that a high discard rate is inefficient. However, the gradient spikes are still clearly present. The final reward ceiling improves only modestly, while the validation reward also becomes more oscillatory, suggesting that simply enlarging the clip ratio does not fundamentally resolve the issue.
> > >
> > > So our point is not only that a high discard rate wastes useful samples, but also that this discard mechanism is tied to the gradient instability we highlighted. Reducing the discard rate through hyperparameter adjustment can partially improve training speed, but it does not eliminate the gradient spikes, which suggests that the underlying sensitivity of sequence-level all-or-nothing clipping remains.
> > >
> > > Thank you again for the constructive feedback and for your support.

---

### Official Review · Reviewer_G4mG · 2026-03-12

**Soundness:** 3
**Presentation:** 3
**Significance:** 3
**Originality:** 3
**Overall Recommendation:** 5
**Confidence:** 3

**Summary:**

This paper analyzes the optimization dilemma of RLVR in the post-training stage of LLMs. Existing token-level methods (such as GRPO) tend to lead to unstable training and large gradient variance, while sequence-level methods (such as GSPO), although it improves stability, will lead to gradient waste caused by over conservative clipping. To this end, the author proposes the TGPO. Core contribution of TGPO is the "dual trust information gate", which dynamically calculates the trust score through the global drift and internal consistency of the sequence. According to the degree of trust, the mechanism makes a smooth transition between sequence anchor and token-level fine-grained update, thus utilizing the information in the imperfect trajectory while maintaining the optimization stability, and improving the sample efficiency.

**Compliance With Llm Reviewing Policy:**

Affirmed.

**Key Questions For Authors:**

1. The calculation of trust score depends heavily on the clipping parameter of Formula 7. The article does not explain why such clipping is necessary or how to select parameters. In addition, such clipping forcibly smoothes the gradient contribution of some high probability tokens. Intuitively, no matter in mathematical tasks or code tasks, there are some "decisive key steps" that may need to obtain a larger gradient.
2. As an optimization method for RL based post training, the author's experiments mainly focus on mathematical reasoning tasks, which often have clear solutions. Is there any data on other tasks that can better reflect the generalization of this method? Is it still possible to effectively use "trust score" for guidance? (such as code task)
3. How is the failure track used? In 5.3, the author uses part of the sequence information of the failure track for gradient update. How are these "partially informational sequences" selected? If the error track accounts for a high proportion, just reducing the weight may not be enough to prevent the model from learning some error patterns. Under large-scale training, will these noises have a long-term impact on the model?

**Limitations:**

Yes.

**Strengths And Weaknesses:**

Strength:
The author focuses on the trade-off between stability and data efficiency in RLVR training, which reveals that relying solely on sequence-level tailoring will lead to the waste of a large number of high-quality gradients. The trust-gating mechanism introduced is reasonably designed. By combining global drift and internal consistency as control signals instead of fixed parameters, the continuous adaptive interpolation of the two is realized. This dynamic adjustment strategy is logically self-consistent, and does not need to introduce additional signals and computational overhead.


Weakness:
1. The theoretical support is slightly insufficient. The paper mainly relies on empirical analysis and experiments. Although trust gating is intuitively effective, there is a lack of strict theoretical analysis to explain why such a form of gating function is adopted, and the selection of some super parameters seems to rely more on experimental tuning, but there is a lack of sufficient theoretical explanation (such as the clipping parameter  \epsilon and  \tao of Formula 12).
2. The definition of "drift" is a little simple. The author assumes that "the smaller the global offset, the more trustworthy the model will be", which is not necessarily true in reinforcement learning. For example, sometimes the model has a large offset (mu is large) on some tracks through exploration because it has found an action track that is better than the old strategy (reward is high). In this case, it will be pulled back by the gating mechanism and reduce the trust. Therefore, the model does not dare to update enough parameters on the higher reward path. This may be more common in code tasks.

---

> ### Author Rebuttal · Authors · 2026-03-29
>
> # 1. Theoretical Support of TGPO
>
> We thank the reviewer for the concern about the theoretical grounding of TGPO’s trust gating. Below we clarify it from a trust-region perspective.
>
> ### (1) Gating = second-moment trust-region proxy
>
> Let $\delta_t = \log \tilde{\rho}_t$, with
>
> $$
> \mu = \frac{1}{L}\sum_t \delta_t,\quad
> \sigma^2 = \frac{1}{L}\sum_t (\delta_t - \mu)^2.
> $$
>
> Then
>
> $$
> \mu^2 + \sigma^2 = \frac{1}{L}\sum_t \delta_t^2.
> $$
>
> So the trust score is
>
> $$
> G = \exp\left(-\frac{\mu^2+\sigma^2}{2\epsilon^2}\right)
> = \exp\left(-\frac{1}{2\epsilon^2}\cdot \frac{1}{L}\sum_t (\log \tilde{\rho}_t)^2\right).
> $$
>
> Thus, $G$ is an exponential penalty on the empirical second moment of log-ratios, i.e., the same second-order quantity governing local policy deviation in trust-region / Fisher-metric methods. Hence, $G$ can be viewed as a soft trust-region scaling factor: larger local shifts receive exponentially smaller trust.
>
> ### (2) Drift + consistency = complete second-order statistic
>
> The decomposition
>
> $$
> G = G_{\text{drift}} G_{\text{cons}}
> $$
>
> is not arbitrary:
>
> * $G_{\text{drift}}$ captures the mean shift ($\mu$);
> * $G_{\text{cons}}$ captures the dispersion ($\sigma^2$).
>
> Together they reconstruct the second moment, separating two failure modes of token-level IS:
>
> * large global shift;
> * high intra-sequence variance (outlier tokens).
>
> ### (3) Winsorization = bounded ratio control without token exclusion
>
> We winsorize the token log-ratio as
>
> $$
> \log \tilde{\rho}_t \in [\log(1-\epsilon),\ \log(1+\epsilon)],
> $$
>
> equivalently bounding the effective ratio by
>
> $$
> \tilde{\rho}_t \in [1-\epsilon,\ 1+\epsilon].
> $$
>
> This is fundamentally different from PPO-style clipping. PPO clips the objective and lose useful gradient. In TGPO, winsorization is applied before constructing the trust-gated mixing coefficient and anchored token weight:
>
> $$
> \mu,\sigma^2 \rightarrow G \rightarrow \alpha,\qquad
> \log w_t=(1-\alpha)\mu+\alpha\log\rho_t.
> $$
>
> Therefore, $\epsilon$ does not truncate token updates. Instead, it bounds the magnitude of token-level correction through ratio statistics. In this sense, $\epsilon$ is the locality radius of a robust trust-region estimator: it prevents a few extreme ratios from dominating the update, without PPO’s “discard outside the interval” behavior.
>
> This matches our principle of *retain rather than exclude*: every token can still contribute, but with controlled strength.
>
> ### (4) $\tau$ = trust sensitivity (upper bound on token-level dominance)
>
> The mixing coefficient
>
> $$
> \alpha = \frac{G}{\tau}
> $$
>
> controls interpolation between sequence-level and token-level updates.
>
> Since $G \in (0,1]$, choosing $\tau=2$ ensures
>
> $$
> \alpha \le 0.5,
> $$
>
> i.e., token-level correction remains bounded and cannot dominate even under high confidence. This enforces a conservative trust-region regime and avoids collapse to high-variance token-wise updates.
>
> # 2 On clipping
>
> Clipping in TGPO is applied to the log-ratio for robust trust estimation, not to the objective itself. Its role is to prevent a few extreme token ratios from dominating the second-moment statistic. We use the standard PPO scale to bound the ratio, not to truncate gradients. TGPO does not discard token updates: all tokens still contribute through the final PPO-style objective. Thus, clipping controls the magnitude of token-level correction rather than suppressing decisive tokens. The improved Pass@k/Avg@k also suggests that key reasoning steps are not materially hindered.
>
> # 3 On the potential risk of drift term
>
> We agree that large drift can correspond to better trajectories. In TGPO, drift is not used to reject trajectories, but only to regulate how much we trust token-level importance weighting. When \(\mu\) is large, the update shifts toward the sequence-level anchor instead of relying on potentially unstable token-wise ratios. High-reward trajectories are still learned, but in a more stable, sequence-level manner.
>
> # 4  Generalization beyond math tasks.
> As a preliminary signal of generalization, we evaluate on a reasoning-intensive QA benchmark(GPQA Diamond) using the same trained models:
>
> | Method | GPQA Diamond |
> |-|-|
> | Base Model | 24.7 |
> | GRPO | 32.3 |
> | GSPO | 34.1 |
> | DAPO | 34.5 |
> | TGPO | 36.2 |
>
> TGPO consistently outperforms prior methods, suggesting that the trust-score mechanism is not limited to math reasoning and can generalize to more open-ended tasks.
>
> # 5 On failure trajectories.
> TGPO does not discard failure trajectories. Low trust reduces (\alpha), so the update falls back toward the sequence-level anchor, weakly using those trajectories while attenuating unreliable token patterns.
>
> The “partially informational sequences” are thus not intentionally selected, but emerge automatically from trust-gated weighting. Training dynamics suggest this does not cause harmful long-term effects: TGPO keeps higher entropy longer and gradually increases trust, enabling broader early exploration without letting noisy failures dominate.

---

> > ### Author Rebuttal · Reviewer_G4mG · 2026-04-02
> >
> > The author answered the design and role of clipping, \episilon and \tau in detail, which enhanced the theoretical motivation and explanation of their idea. However, the theoretical motivation for parameter selection was not fully explained, and the influence of key parameters on the results was not shown. I still believe that the parameter values selected in the paper highly depend on experimental tuning.
> > In task promotion, the author gives an additional result of reasoning intensive QA, which can explain the generalization of the method to a certain extent. But I mentioned the code task twice in my questions, not to set it as the only additional necessary verification scenario, but because code tasks, unlike mathematical tasks with unique correct answers, yet unlike natural language QA with considerable semantic flexibility, are more likely to expose the “high-value but high-drift” issue in policy optimization. when large drift arises from better exploration and corresponds to higher rewards, TGPO does not discard such trajectories (as the author explains, it degrades to more conservative sequence-level learning). However, by reducing the trust weight, might it weaken updates that should be strongly reinforced, leading to systematic under-updates for high-value novel trajectories? In this regard, the newly added reasoning-intensive QA results do show some generalization, but they are insufficient to directly address the above concern. At least for me, they do not fully alleviate the worry that such “more conservative” handling may suppress high-value exploration.
> > On the issue of failed trajectory handling, the author has directly explained how TGPO processes incorrect trajectories (via automatic trust-gated weighting to determine updates), and claims no long-term harmful effects were observed in the training dynamics.

---

> > > ### Author Response · Authors · 2026-04-04
> > >
> > > We thank the reviewer for the thoughtful follow-up. We respond to the concerns on $\epsilon$, $\tau$, and the code-task setting below.
> > >
> > > Regarding $\epsilon$, our choice is not based on tuning. As discussed in our derivation, letting $\delta_t=\log \tilde{\rho}_t$, we have
> > >
> > > $$
> > > \mu=\frac{1}{L}\sum_t \delta_t,\qquad
> > > \sigma^2=\frac{1}{L}\sum_t (\delta_t-\mu)^2,
> > > $$
> > >
> > > and thus
> > >
> > > $$
> > > \mu^2+\sigma^2=\frac{1}{L}\sum_t \delta_t^2.
> > > $$
> > >
> > > So Eq. (10) can be rewritten as
> > >
> > > $$
> > > G=\exp\!\left(-\frac{\mu^2+\sigma^2}{2\epsilon^2}\right)
> > > =\exp\!\left(-\frac{1}{2\epsilon^2}\cdot \frac{1}{L}\sum_t (\log \tilde{\rho}_t)^2\right).
> > > $$
> > >
> > > This makes the role of $\epsilon$ clear: it is the locality scale used to normalize the empirical second moment of the winsorized log-ratios. In other words, $\epsilon$ determines what magnitude of local policy deviation should still be treated as trustworthy. This is also why it is naturally tied to Eq. (7), where we winsorize
> > >
> > > $$
> > > \log \tilde{\rho}_t \in [\log(1-\epsilon),\ \log(1+\epsilon)].
> > > $$
> > >
> > > So $\epsilon$ is not an extra free parameter introduced only for the gate; it is the same radius that governs both robust trust estimation and the trust-region-like normalization. Dividing by $\epsilon^2$ makes the statistic dimensionless relative to this radius, and the factor $1/2$ follows the standard quadratic form used in second-order KL / trust-region approximations. Importantly, in all our experiments we simply used the standard PPO clipping value $\epsilon=0.2$ directly.
> > >
> > > For $\tau$, we agree that there is some degree of parameter choice. However, the sensitivity is mild. As shown in Section 4.3, Table 3, using $\tau=1$ or $\tau=3$ leads to only small drops relative to $\tau=2$. We selected $\tau=2$ mainly because it gives the most natural interpretation: in the high-trust regime, since $\alpha=G/\tau$ and $G\to 1$, we have $\alpha\to 1/2$, and Eq. (13) becomes
> > >
> > > $$
> > > w_t = S^{1-\alpha}\rho_t^\alpha \approx S^{1/2}\rho_t^{1/2},
> > > $$
> > >
> > > namely a balanced geometric mean between the sequence-level ratio and the token-level ratio. So the final choice was not based on heavy tuning, but mainly on robustness plus interpretability.
> > >
> > > On the reviewer’s concern about code tasks and the “high-value but high-drift” issue, we agree that this is an especially meaningful setting. Code is indeed different from both mathematical tasks with unique answers and open-ended QA with greater semantic flexibility: it is structured, execution-sensitive, and more likely to expose cases where large drift may come from genuinely better exploration. Because these experiments required additional training, we were only able to complete them slightly later, and we thank the reviewer for pushing us in this direction.
> > >
> > > We conducted a quick additional test on code tasks using Qwen2.5-Coder-7B-Instruct, trained on the code split of the PRIME RL dataset, and evaluated on LeetCode and LiveCodeBench. The greedy pass@1 results are:
> > >
> > > | Method | LeetCode | LiveCodeBench |
> > > |---|---|---|
> > > | Qwen2.5-Coder-7B-Instruct | 50.6 | 34.3 |
> > > | GRPO | 60.0 | 35.4 |
> > > | GSPO | 55.7 | 34.9 |
> > > | TGPO | 62.5 | 35.6 |
> > >
> > > We observe a consistent improvement here as well. Moreover, the training dynamics remain qualitatively similar to what we reported in the paper: the trust score still correlates approximately linearly with reward, and entropy does not collapse prematurely. A plausible interpretation is that novel code solutions, whether correct or incorrect, often induce larger drift and thus lower trust. Under TGPO, such trajectories are not discarded, but receive smaller and more conservative probability updates, which helps preserve exploration space rather than prematurely over-reinforcing brittle local patterns.
> > >
> > > Code also highlights an interesting future direction. Compared with mathematical reasoning, code generation may place stronger demands on variable-aware or execution-aware credit assignment, and may also benefit from richer mechanisms for diversity and exploration shaping. TGPO mainly addresses instability and gradient waste caused by unreliable token-level weighting; combining it with more code-specific credit-assignment or exploration strategies is a promising direction for future work.
> > >
> > > We hope these clarifications and additional results help address the reviewer’s concern.

---

### Decision · Program_Chairs · 2026-04-30

**Decision:**

Accept (regular)

**Comment:**

This paper introduces Trust-Gated Policy Optimization (TGPO), a framework designed to address the stability-efficiency tradeoff in RLVR by adaptively interpolating between token-level and sequence-level updates. The reviewers reached a consensus on the practical importance of the problem and the intuitive appeal of the "retain rather than exclude" strategy. Key concerns initially focused on the lack of theoretical justification for the Gaussian-based trust score and the potential for the "drift" term to suppress high-value exploration, particularly in code-related tasks. The authors provided a technically rigorous rebuttal that addressed these doubts. While Reviewer fymj maintained a rejection, it was based on the structural incompleteness of the original manuscript rather than unresolved technical flaws. The authors must holistically integrate the theoretical derivations and the code-task results into the subsequent version to resolve the remaining concerns about the manuscript's completeness.